# Classification of Shredded Aluminium Scrap Metal Using Magnetic Induction Spectroscopy

**DOI:** 10.3390/s23187837

**Published:** 2023-09-12

**Authors:** Kane C. Williams, Michael J. Mallaburn, Martin Gagola, Michael D. O’Toole, Rob Jones, Anthony J. Peyton

**Affiliations:** 1Department of Electrical and Electroninc Engeering, The University of Manchester, Oxford Road, Manchester M13 9PL, UK; michael.mallaburn@manchester.ac.uk (M.J.M.); michael.otoole@manchester.ac.uk (M.D.O.); a.peyton@manchester.ac.uk (A.J.P.); 2Magnapower Equipment Ltd., A1, Harris Business Park, Hanbury Rd., Stoke Prior, Bromsgrove B60 4FG, UK

**Keywords:** classification, electromagnetic induction, magnetic induction spectroscopy, recycling, waste recovery, industrial conveyor, Twitch waste stream

## Abstract

Recycling aluminium is essential for a circular economy, reducing the energy required and greenhouse gas emissions compared to extraction from virgin ore. A ‘Twitch’ waste stream is a mix of shredded wrought and cast aluminium. Wrought must be separated before recycling to prevent contamination from the impurities present in the cast. In this paper, we demonstrate magnetic induction spectroscopy (MIS) to classify wrought from cast aluminium. MIS measures the scattering of an oscillating magnetic field to characterise a material. The conductivity difference between cast and wrought makes it a promising choice for MIS. We first show how wrought can be classified on a laboratory system with 89.66% recovery and 94.96% purity. We then implement the first industrial MIS material recovery solution for sorting Twitch, combining our sensors with a commercial-scale separator system. The industrial system did not reflect the laboratory results. The analysis found three areas of reduced performance: (1) metal pieces correctly classified by one sensor were misclassified by adjacent sensors that only captured part of the metal; (2) the metal surface facing the sensor can produce different classification results; and (3) the choice of machine learning algorithm is significant with artificial neural networks producing the best results on unseen data.

## 1. Introduction

As countries move toward a circular economy, we face multiple challenges to reach this goal. A significant challenge is metal recycling, which needs improvement to allow for a sustainable economy [1]. A main advantage of recycling is that it reduces the need to use virgin ore, which is finite. Additionally, the energy required can be less when using recycled (secondary) ore than virgin (primary) ore. Recycled aluminium is an excellent example, as for 1000 kg of aluminium produced, 11,690 MJ of energy is needed using recycled aluminium, whereas virgin bauxite ore requires 186,262 MJ [2]. Primary aluminium, in general, is energy-intensive as, when compared to steel, it requires 8–9 times the energy [3,4] to produce.

It has been reported there will be a surplus of aluminium scrap over the next decade if no technology and supporting strategies are developed [5]. As well as a surplus, there will be an increased demand for wrought aluminium for vehicles as aluminium continues to be used for vehicle lightweighting [3]. If we cannot effectively recycle the aluminium scrap, we must create more primary aluminium, resulting in higher carbon emissions and reduced economical value [5]. These factors make it vital for new technology to be developed that can economically and accurately separate large quantities of scrap aluminium.

Recycling aluminium and aluminium alloys present several key challenges. For instance, tramp elements are present within aluminium alloys. Tramp elements are allowed at a pre-specified rate; however, if this rate is exceeded, it can change the metal’s properties, such as making the metal more brittle [6]. The tramp element rate makes it vital not to mix aluminium alloys whose secondary element is different, as this would lead to a change in metal properties. Tramp elements can be removed from some scrap metal, but the energy required to achieve this must be considered [6]. However, aluminium’s thermodynamic barriers make it difficult to remove tramp elements [6].

Aluminium that needs to be recycled can be grouped into two main groups, cast and wrought. Wrought aluminium has been processed through mechanical methods like rolling, whereas cast is aluminium poured into a mould. When recycling cast aluminium, wrought aluminium can be present in the mix, as cast aluminium is more resilient to impurities [3]. However, no cast aluminium can be present within the wrought aluminium mix. It was common for cast and wrought aluminium to be processed into cast aluminium for items like engine blocks, but this has become less economical [3]. Current methods aim to extract wrought aluminium from cast aluminium to make aluminium recycling. The mixture of the cast and wrought aluminium waste stream is called Twitch, defined as “floated fragmentiser aluminium scrap from Automobile Shredders” [7].

Twitch is acquired from an initial waste stream consisting of ferrous, non-ferrous metals and non-metals such as steel, copper, aluminium, rubber and plastics [8]. This mixed waste stream would first pass through magnets, removing all ferrous metals like steel and iron. The remaining mix would pass through an eddy-current separator (ECS) to remove all non-ferrous metals. The extracted non-ferrous metals would need to be further separated to remove the aluminium to create Twitch.

A common approach to sorting the non-ferrous scrap is to use the sink–float method to separate the metals based on their density. The sink–float method uses different gravitational drums, which consist of water and heavy media such as ferrosilicon or magnetite, to separate the metals [9]. The sink–float method separates the light aluminium from the heavier copper and brass. Another approach, similar to sink–float, is manual sorting, where a worker separates the metal based on its colour, weight and texture [9]. The accuracy of manual sorting depends on the worker’s experience.

Once Twitch has been extracted, the mixed aluminium must be further separated into cast and wrought aluminium. A commercial method to extract wrought is Laser-Induced Breakdown Spectroscopy (LIBS), which uses a laser to ablate the surface of the metal; this causes a plasma that can be analysed to determine its composition [10]. To improve LIBS, the addition of a 3D camera has been proposed to detect the sample and the optimal target flat and uncontaminated surface [10]. Another commercial method uses X-ray fluorescence (XRF) to determine the metal composition [11]. However, it can be difficult for XRF to classify aluminium alloys [9]. XRF uses the majoring alloy element to determine the spectral ratio for aluminium alloys [9]. Both LIBS and XRF are sensitive to surface contamination, which would require the metal to be cleaned prior, increasing the cost and time.

LIBS and XRF are expensive methods and difficult to scale, so it can be economical to reduce the quantity of Twitch passing through them. Air knives/air separation [8] could be used prior, which blows air to drop the effectively heavier lumps of cast aluminium. Large wrought aluminium pieces may drop as they are too heavy, and smaller cast pieces will be light so that they will be ejected. As wrought aluminium is the target metal, reducing the quantity of cast aluminium is preferred.

Separation could be achieved by hand as the weight of wrought tends to be lighter than cast, and the texture of the two can be visually differentiated. However, hand sorting is only economical when labour costs are low. Díaz-Romero et al. [12] used the visual difference between cast and wrought aluminium to train a deep-learning algorithm to separate the metal. The algorithm was able to obtain a 0.98 F1 score that used RGB and the depth of the sample of the 82 test metal samples [12]. Surface contamination on the metal samples can affect vision systems, which may require prior cleaning, which increases the cost and time.

More cost-efficient solutions are currently implemented in research which use the conductivity of the metal sample to differentiate them. The difference in conductivity between cast and wrought aluminium was highlighted as a benefit in previous studies when extracting wrought from cast [13]. The metal’s conductivity is reported as a percentage relative to the conductivity of copper, which is called the International Annealed Copper Standard (ICAS). The conductivity of the metals is 23% ICAS for cast and >40% ICAS for wrought [14]. The difference in conductivity is used to separate the metals within this study.

Electrodynamic sorting technology was developed at the University of Utah, which implements a tuneable or variable frequency ECS [15,16]. The authors demonstrated that this technology could separate Al-6061 from AL-2014 aluminium [15], though this was with metal spheres. Electrodynamic sorting was able to remove aluminium from Zorba, with 97.6% purity and 93% recovery. The system could not eject small pieces of aluminium due to the excitation frequency, which reduced the recovery rate [17].

We propose using magnetic induction spectroscopy (MIS) to classify wrought aluminium from cast aluminium based on their conductivity difference. MIS excites multiple frequency components across a spectrum, whereas the electrodynamics sorting technology uses one excitation frequency. A benefit of using multiple frequencies is that it allows us to account better for the shape effects on the measurements, which allows the measurements to represent the conductivity of the metal sample better. Unlike electrodynamics sorting technology, MIS needs to have a separate ejection system to separate the metals, such as air ejectors. A benefit of MIS over LIBS and XRF is its lower price, and it is not affected by surface contamination. Ideally, MIS would be used to remove wrought aluminium from cast aluminium. However, it could also be used to reduce the quantity of cast aluminium in the waste stream prior to other methods, similarly to air knives.

This paper first discusses results using a test rig first introduced in previous work [13,18] to highlight the classification performance, where we use a larger dataset for the waste stream than in previous work. We then implement the MIS sensors on an industrial conveyor travelling at 2 m/s to train and test a machine learning algorithm as our first industrial trial. The algorithm will detect and classify the metal, and then the output will determine whether the air ejectors should fire, pushing the metal into a separate bin. Finally, we discuss the improvements needed to allow MIS to work as part of an industrial system at speed and further improvements to the machine learning algorithm.

The contributions of this paper are three-fold: Firstly, we show the first use of magnetic induction spectroscopy to classify wrought aluminium from cast aluminium. Secondly, we use a larger dataset for the Twitch waste stream than in previous work. Thirdly, this is the first paper to demonstrate a full MIS-based separation system that implements the sensors and an ejector system on a conveyor belt system at industrial speeds of 2 m/s.

## 2. Theory

To formally define the magnetic induction spectra and understand how MIS can discriminate between metals based on conductivity, we consider the case of a conductive sphere in free space illuminated by a uniform oscillating magnetic field [19]. While simplified, this case provides a well-known and straightforward analytical solution for eddy-current or magnetic scattering problems, the broad relations of which generalise to object shapes beyond a sphere. A conductive non-magnetic sphere centred at the origin is placed within a uniform magnetic field oscillating at a frequency of *f*. The sphere has a radius of *a*, conductivity of σ and permeability of $\mu =\mu_{0}=4\pi \times 10^{-7}$. The magnetic field acts along an axis *Z* and induces eddy-currents within the sphere, which flow in an azimuthal direction [20]. These eddy-currents induce a secondary magnetic field. Taking a point *z* along the *Z* axis outside the sphere (z>a), denote the excitation magnetic field at this, as $H_{ex}\left(t\right)=H_{ex}\left(f\right)e^{j2\pi ft}$. The secondary field emitted by the sphere as $H_{rx}\left(t\right)=H_{rx}\left(f\right)e^{j2\pi ft}$, where $H_{ex}\left(f\right)$ and $H_{rx}\left(f\right)$ are the complex frequency components of each field, respectively. The magnetic induction spectrum, defined as $H_{rx}\left(f\right)/H_{ex}\left(f\right)$, is given by the following,
(1)$$\frac{H_{rx}\left(f\right)}{H_{ex}\left(f\right)}=-\frac{3a^{3}}{z^{3}}\left(\frac{1}{\alpha^{2}}+\frac{1}{3}-\frac{\cosh\alpha }{\alpha \sinh\alpha }\right)$$
$$\alpha ={\left(i2\pi f\sigma \mu \right)}^{\frac{1}{2}}a$$

Figure 1a shows the real and imaginary components of two metallic spheres with a radius of 20 mm at the point *z*, 3 mm from the surface. The conductivity of the spheres is 23% and 44% ICAS to represent cast and wrought aluminium, respectively, [14]. It can be seen in Figure 1a that the conductivity changes the spectra of both the real and imaginary components. The real component starts from zero when f=0 and reduces to a high-frequency asymptote as *f* increases to infinity. The frequency decrease from 0 to the asymptote differs depending on the conductivity, where the higher conductive sphere drops first. A difference in the imaginary component is also observed, where the trough position differs depending on the conductivity. The higher conductivity metal trough occurs at a lower frequency.

Figure 1b shows the real and imaginary components of a cast aluminium sphere with a 20 and 23 mm radius, again at a point *z*, 3 mm from the surface. It can be seen in Figure 1b that when the sphere radius increases, the asymptote of the real component increases, which did not occur when the conductivity varied. The imaginary component trough also increases in size when the radius of the sphere increases, whereas when the conductivity changes, the frequency at which the peak occurred varies and height does not.

From this discussion, it is evident that shape (i.e., radius) and conductivity impact the characteristics of the magnetic induction spectra in different ways and to differing degrees depending on frequency. There are notable features, such as the asymptote of the real component, which, due to the skin-depth effect, is subject to size but independent of conductivity. We propose using the spectra to discriminate between materials of different conductivity while minimising the impact of varied geometry. We are using spheres to illustrate the principles here. However, we apply the same argument to metal fragments of random geometry, using a multi-frequency approach to enhance the conductivity contrast while minimising the effect of shape variation.

## 3. Method

In the following, we determine the ability to use multi-frequency magnetic induction spectroscopy to classify wrought from cast aluminium in the Twitch waste product; a mixed aluminium waste stream recovered from an automobile shredder. We describe two measurement systems: in the first, we evaluate MIS performance using a static test rig with the so-called ’MetalID’ sensor [21]. Here, metal fragments are placed manually on top of the sensor, centrally to the sensor’s axis, to measure an induction spectrum while the fragment is stationary. This method demonstrates the ability of MIS to classify aluminium fragments and presents an upper bound on predicted performance under artificial, controlled, precise and repeatable conditions.

In the second, we describe the industrial MIS system, which is a full material separation solution, adapting the MetalID sensor to operate under a fast-moving (2 m/s) conveyor. The industrial system also combines an air ejector manifold that blows fragments classified as wrought into bins to separate from the falling stream exiting the conveyor.

### 3.1. Static MetalID System

The MetalID sensor forms part of the static test rig for the induction measurements of metal fragments previously described in Williams et al. [13,18]. A schematic of the system is shown in Figure 2. Briefly, it comprises the following components:A sensing element consisting of concentric coils (excite and receive coils) to drive an excitation magnetic field and measure the resultant scattered field. The full details of the sensing element are described in [21].Front–end analogue drive and receive electronics, including power and low-noise instrumentation amplifiers.An STM Nucleo-H7A3ZI-Q board for waveform generation and data acquisition.

The MetalID sensing element has an inner excite coil (32 turns) wrapped around a 10 mm diameter former containing two 6 mm diameter rods (Fair-Rite 4077276011). A 16 mm diameter acetyl former encloses the structure, with two outer receive coils (600 turns) wrapped around. The receive coils are in a gradiometer arrangement to cancel out the effect of excitation. The sensing element is enclosed by an aluminium cylinder for screening, with the top open to allow for the detection of the test pieces.

The Nucleo-H7A3ZI-Q generates a 12-bit DAC excite signal composed of six superimposed waveforms of frequencies 3, 6, 9, 15, 30 and 60 kHz, as in previous work [22]. The excite signal is input to a power amplifier (OPA541AP, Texas Instruments) which drives the excite coil. The received waveform was measured using a low-noise instrumentation amplifier (AD8429, Analog Devices) with the microcontroller internal ADC sampling at 1.6 MSPS, filling a 256-element FFT buffer to obtain individual frequency components. The sensor was calibrated using a 10 mm dia. and 20 mm-long ferrite cylinder (material 4B1, Ferro cube). The ideal ferrite response yields a purely real component uniform across the frequencies. The calibration parameters were saved and processed within the data acquisition device; this allows the device to output the calibrated measurement.

The static test rig also incorporates a camera placed above the MetalID sensor. A full description of the static MetalID system can be found in Williams et al. [13], where an image of each sample was taken along with the induction measurement to extract the fragment colour. In this work, images of the fragments were also taken to identify which pieces were misclassified and how they were presented to the sensor. The static MetalID system measurements were taken by placing a metal sample on the MetalID sensor, centralising the sample above the coils. The average of 16 induction measurements was taken with a matching camera image. Once the measurement was complete, the metal sample was flipped and measured again to present the opposite surface to the sensing element.

### 3.2. Industrial MIS System

The industrial MIS systems progressed the MetalID sensor towards a full material separation solution able to classify and extract target material from the input waste stream at high throughputs. The concept is illustrated in Figure 3. The system consists of MetalID sensors, air ejectors and a conveyor belt.

The MetalID sensors are mounted in an aluminium enclosure and fixed directly under a conveyor belt to ensure the metal pieces’ proximity to the sensors. The enclosure and its internal components are shown in Figure 4. The industrial prototype uses eight sensors to create a sensitive zone of approximately 150 mm across the conveyor. The sensors are positioned to overlap so that a metal sample would be detected by at least one sensor. Each sensor’s data acquisition and front–end electronics included a power amplifier (LT1210, Linear Technologies) and a low-noise differential amplifier (AD8429, Analog devices) mounted in the aluminium enclosure. The Red Pitaya STEM 125-14 data acquisition device was used and connected to two separate sensors, with 4 Red Pitayas used on the industrial MIS system. Networking hardware was also incorporated into the enclosure to communicate with individual Red Pitayas for setup and calibration.

An array of air ejectors were mounted at the discharge end of the conveyor. An ejector extracted target material by using blasts of compressed air to push metal samples into the furthest bin as they leave the conveyor, thus separating selected samples from other metals allowed to free-fall, as shown in Figure 3. The system had 40 air ejectors in total, spaced 15.75 mm apart. However, only eight ejectors were used in this study, which were aligned with each MetalID sensor. The air ejectors and Red Pitayas were connected to an industrial PLC, which allowed the Red Pitayas to trigger the air ejectors after a predefined delay set by the PLC.

An industrial conveyor belt system, shown in Figure 5, was used. The industrial conveyor belt was 600 mm wide, 2 m long and 1.3 mm thick. A vibratory feeder was used to deposit metal samples onto the conveyor, which is common industry practice. Plastic guides were attached to the vibratory feeder and above the belt to ensure the metal samples passed over the sensor array. Figure 6 shows the position of the feeder, guides, and sensors on the conveyor belt. The conveyor speed was set to 2 m/s to match typical throughputs used in the industry.

For experiments, the metal samples were first dropped onto the conveyor of the industrial MIS system. The samples then travelled at 2 m/s until they passed over the MetalID sensor, which would detect the sample and start recording measurements. A machine learning model, described later in Section 3.4, is implemented on the Red Pitaya to classify the samples. The output of the machine learning model is binary, where ‘1’ triggers a GPIO pin connected to the PLC. After a prescribed delay, the PLC triggers the air ejector in the same position as the MetalID sensor as the sample reaches the end of the conveyor. For this study, the machine learning model was trained to trigger when it detected wrought aluminium and let any cast aluminium drop.

The sensors were calibrated with the static MetalID system using ferrite targets. In this case, a 10 mm-diameter 200 mm-long ferrite rod was attached to the conveyor belt along the length of the sensitive zone and moved at 2 m/s. Multiple calibration measurements were taken of the rod and averaged. A measurement threshold was set using the magnitude summed over the six frequency components to detect when a metal piece was present above the sensor and start recording. The recorded profile of each sample pass was then condensed into a single complex number for each frequency by taking the peak measurement. These six complex frequency values were used as input features for the machine-learning model.

The sample sets are described in Section 3.3. Training samples used to develop the machine learning model were deposited on the conveyor using the vibratory feeder, whereas the test samples were dropped one at a time by hand for a more controlled data collection process. We recorded when the air ejectors fired, which indicates the machine learning output. This ensured our results represented the actual classification performance of the sensor and removed any impact from sample misdirection or mistiming from the air ejectors.

Measurements taken for training were stored on the Red Pitaya and extracted once all measurements were complete. These measurements were then used to train the machine-learning model on a PC. The machine-learning model was then implemented on the Red Pitaya with model parameters imported from the training PC. The Red Pitaya would then provide real-time measurement, processing, and classification of each test sample as it passes the MetalID sensor.

### 3.3. Test Samples

Four datasets were used, which consisted of different ratios of wrought and cast aluminium sourced from a Twitch [7] commercial waste stream. The typical ratio of Twitch waste stream is roughly 20% wrought and 80% cast aluminium mix. The proportion of pieces present in each dataset is shown in Table 1.

The dataset ‘Static-training’ consisted of 192 twitch samples with a ratio of 45%/55% (wrought/cast), providing a reasonably balanced dataset to train and test our machine learning algorithm. A balanced dataset was used to return performance metrics free from input material bias and reduce the algorithm’s potential overfitting to the more dominant metal. The ‘Static-training’ dataset will be used on the static MetalID system and later on the industrial MIS system to train a classification algorithm.

The dataset ’Industrial-test’ consisted of 77 twitch samples with a ratio of 53%/47% (wrought/cast). This will be used for testing on the industrial MIS system, providing an unseen and balanced dataset to evaluate the performance of the machine learning algorithm trained on the ‘Static-training’ dataset.

The dataset ‘Industrial-twitch’ consisted of 263 samples with a ratio of 22%/78% (wrought/cast) from a Twitch waste stream which were not processed into a balanced ratio but represented a typical industrial ratio. ‘Industrial-twitch’ was used for testing on the industrial MIS system.

The dataset ‘Industrial-air knives’ consisting of 127 Twitch samples with a ratio of 40%/60% (wrought/cast), which is created by passing the ‘Industrial-twitch’ dataset through an air knives system. Air knives systems are a method that could be used to reduce the quantity of cast prior to the use of more precision technologies such as XRF and LIBS. ’Industrial-air knives’ is therefore representative of a material recovery process featuring an air knives pre-sorting stage.

To create ‘Industrial-air knives’, the pieces from the ‘Industrial-twitch’ dataset would first pass along a vibratory feeder until they reached the end, where air was blown out of nozzles. Ideally, the air blows the wrought pieces, whereas the cast would be allowed to drop freely under gravity. Two bins were set up, with an adjustable bin separator used to determine how much-blown material falls into the furthest bin. Smaller cast pieces tended to be blown away with the wrought, and large wrought pieces dropped as there was not enough force to push them past the separator. All samples thrown into the furthest bin are grouped into the ‘Industrial-air knives’ dataset. The air knives dropped 127 (1470 g) pieces of cast and 9 pieces (119 g) of wrought. The ‘Industrial-air knives’ dataset was used to evaluate the performance of a combined system that used air knives and an industrial MIS system.

The results are reported in terms of the number of samples and mass for the ‘Industrial-twitch’ and ‘Industrial-air knives’ datasets, the latter in line with industry preference. Apart from ‘Industrial-air knives’, all datasets were visually inspected and weighed by hand before manually separating the two classes of wrought and cast.

### 3.4. Machine Learning

The machine learning methods used in this study are more traditional algorithms for consistency with the approach from our previous work [13]. The algorithms were chosen as they are well established, and their structure and decision process can be visualised. However, in this work, we also include a ’black box’ approach using neural networks for classification. The algorithms used, and the hyperparameters changed during cross-validation, can be found in Table 2.

Other work has compared the classification results of traditional algorithms such as random forests and ANN for the high-resolution prediction of building energy consumption, where it was reported that ANN performed only marginally better than the decision tree-based algorithm [23]. Ren compared ANN and SVM and found that both algorithms had a 10% improvement on the F1 score in testing when the datasets were balanced, but ANN performed better when the dataset was not balanced [24]. When SVM, RF, and ANN were compared to classify tree species from airborne hyperspectral APEX images, it was found that ANN performed better than SVM and RF [25]. It was reported that the benefit of using SVM and RF was the quicker training time, and their simpler methods allowed easier repeatability to produce higher accuracies [25].

The Python library TensorFlow V2.11.0 [26] was used to train and implement the artificial neural network algorithm. Scikit-learn V1.2.1 [27] was used to train and implement all other algorithms. SVM, KNN and ANN input data were scaled between 0 and 1 using Scikit-learn’s MinMaxScaler function. Scikit-learn’s RandomizedSearchCV function was used for ANN, and the GridSearchCV function was used for all other algorithms to determine the best hyperparameters.

The architecture of the ANN used to evaluate unseen pieces is seen in Figure 7. The model consisted of three dense layers of 64 neurons, with a batch normalisation layer prior. The early stopping function from Tensorflow was implemented to reduce overfitting by monitoring the loss of the model. If the loss of the model stopped improving, then training would stop. When the ANN was used with cross-validation, the same architecture was used with varying hidden dense layers. The model outputs a single neuron which uses the sigmoid activation function. The model was trained using Adam with Nesterov momentum [28] as the optimiser, with 20% of the training dataset for validation when tuning the neural network hyperparameters.

### 3.5. Analysis and Comparison

The F1 score is the primary metric used to compare the performance of the different algorithms. It is the harmonic mean between precision and recall [29]. A benefit of the F1 score is that it is only high if the precision and recall are high [29]. The precision and recall are also reported. In what follows, we will refer to precision and recall as purity and recovery, respectively. The purity and recovery rate is the standard terminology for the recycling industry. The F1 score, purity and recovery rate are formally defined using the following equations: $$\mathrm{Purity}=\frac{\mathrm{TP}}{\mathrm{TP}+\mathrm{FP}},\;\mathrm{Recovery}=\frac{\mathrm{TP}}{\mathrm{TP}+\mathrm{FN}}$$
$$F1=2\times \frac{\mathrm{purity}\times \mathrm{recovery}}{\mathrm{purity}+\mathrm{recovery}}=\frac{\mathrm{TP}}{\mathrm{TP}+\frac{\mathrm{FN}+\mathrm{FP}}{2}}$$
where TP, TN, FP and FN are true positives, true negatives, false positives and false negatives, respectively.

In the first set of experiments, the static MetalID system of Section 3.1 was used, measuring each sample of the dataset ‘Static-training’. The algorithms are evaluated using stratified 10-fold cross-validation to keep each class’s ratio within the folds. Cross-validation splits the input dataset into predefined folds, then trains with all the data except one reserved fold for testing. This process is repeated until all folds have been evaluated as a test set. The dataset was shuffled prior to cross-validation, which was repeated a total of 10 times. Each shuffle changed the order of the input data, but the order of the shuffled data was identical between different algorithms to allow a fair comparison. From this analysis, the highest performing algorithm is selected for implementation on the industrial MIS system, using the hyperparameters for that algorithm found to yield that performance.

In the second set of experiments, the industrial MIS system of Section 3.2 was used with only one machine learning model implemented, chosen by the previous experiments as described. In this case, discrete training and test sets were used rather than cross-validation. The dataset ‘Static-training’ was used for training. The samples of that dataset were deposited on the conveyor belt using a vibratory feeder. The industrial MIS system was set to a record mode, storing each sample’s six complex frequency components as they passed over, which were then retrieved from each Red Pitaya after the samples cleared the conveyor. These data were then used to design a machine-learning model uploaded onto each Red Pitaya. The datasets ‘Industrial-test’, ‘Industrial-air knives thrown’, and ‘Industrial-air knives dropped’ were used for testing. Here, the samples were fed on the conveyor sample-by-sample with the classification result noted by the air ejector firing as described in Section 3.2.

When training the models to evaluate their performance for Section 4.3, the SVM algorithm was trained with the data once using the training data with the best hyperparameters obtained from Section 4.1. The models were then evaluated on the test data. As the weights and biases are randomly initialised for ANN, the training and testing sequence was repeated 40 times to give a representative result and allow the calculation of the average F1 score, recovery and purity. The best ANN model was also reported, which was determined by the model that achieved the highest F1 score with the test data. All the machine learning model’s training data were shuffled in the same order to allow a fair comparison.

## 4. Results and Discussion

In the first part of this section, we explore the results of different machine learning algorithms to classify wrought from cast aluminium using the static test rig described in Section 3.1. The static MetalID system provides the most ideal, controlled, and repeatable conditions to measure the scrap metal, presenting a likely upper bound on expected performance and feasibility. These conditions include the metal sample positioned central to the coil and the absence of variations, noise or artefacts caused by sample movement on a conveyor belt.

In the second part, we use the newly developed industrial MIS system to explore the practical performance of this technology in a complete waste sorting solution, where previously controlled factors on the static test rig cannot be easily controlled. Overall, the industrial MIS system did not perform as well as the static MetalID system. Section 4.2 and Section 4.3 will discuss the reasons for this difference evidenced by follow-up experiments.

### 4.1. Static MetalID System

The static MetalID system was used to take six-point induction spectral measurements of stationary metal samples under ideal conditions. The measurements were taken with the ‘Static-training’ dataset, which were used to determine the machine-learning model and hyperparameters that give the highest F1 score using stratified 10-fold cross-validation. The model with the highest F1 score was then chosen for implementation on the industrial MIS system, with results described in the subsequent section.

Figure 8 shows F1 score, recovery, and purity rates of each machine-learning model used. It is found that SVM, LDA and ANN algorithms achieved the highest F1 score, with SVM achieving the highest overall score of 0.9222. RF, ET and Adaboost achieved similar F1 scores between 0.8165 and 0.8236, with recovery and purity rates between 79 and 81% and 83–85%, respectively. KNN performed the worst, with an F1 score of 0.7565. All algorithms except for KNN achieved a purity rate >83%, with SVM achieving the highest with 94.96%. The recovery rate was lower for all algorithms. Apart from KNN, all algorithms achieved a recovery rate of >79%, with SVM again achieving the highest at 89.66%.

SVM has consistently shown the highest performance across all three metrics (F1 score, purity, and recovery rate) and is therefore selected for testing with the industrial MIS system in the work that follows.

### 4.2. Industrial MIS System

The samples in the dataset ‘Static-training’ were measured on the industrial MIS system and used for training the SVM algorithm. Recall that the ‘wrought’ class should trigger the air ejector causing the piece to eject into a bin, whereas the ‘cast’ samples drop. The datasets ‘Industrial-test’, ‘Industrial-air knives’, and ‘Industrial-twitch’ were then used for testing following the methods described in Section 3.2 and Section 3.5. Each dataset represents a different class ratio of wrought and cast aluminium. Figure 9 shows the recovery and purity rates achieved with these datasets. The best result from Section 4.1 is also shown for comparison.

The ‘Industrial-test’ dataset is balanced with a class ratio of 47%/53% (wrought/cast) to determine the performance independent of the sample prevalence in the input material. In this case, the trained machine learning model achieved a 63.89% recovery and a 60% purity rate, a reduction from the 89.66% and 94.96% achieved on the static MetalID system, which used a similar ratio.

As described in Section 1 and Section 3.2, a pre-sorting stage using air knives before sensor-based sorting could improve the balance ratio between the metals. Essentially, a blade of air pushes metal fragments as they fall, separating them according to their weight and the air resistance they present. This process tends to capture more wrought aluminium, which presents as less aerodynamic pieces than the lumps of cast, increasing its proportion in the mix compared to the typical Twitch composition.

The dataset ‘Industrial-air knives’ was processed using commercial air knives, yielding a ratio of 40%/60% (wrought/cast). The recovery and purity rate for this dataset falls to 52.94%. Industry preference often reports performance in terms of the mass of material rather than sample numbers. Therefore, of the 836 g material in the ‘Industrial-air knives’ dataset, 123 g of wrought aluminium is extracted and mixed with 127 g of cast, which gives a purity rate of 49%, or a resultant 49%/51% (wrought/cast) ratio of the output product. This leaves 138 g of wrought not recovered, yielding a recovery rate of 47%.

The third dataset, ‘Industrial-twitch’, has a composition typical of a Twitch waste stream with a ratio of 23%/77% (wrought/cast) aluminium when using the number of samples. As might be expected from the falling prevalence of wrought in the fraction, the performance of the industrial MIS system further diminishes. We find a recovery rate of 51.67% and a purity rate of 36.90%. The initial twitch ratio by mass was 16%/84% (wrought/cast), with a total mass of 2425 g. The recovery and purity rates were 52.62% and 32.65%, respectively, with 191 g of wrought successfully extracted and the remaining 189 g not recovered.

We further compare how the industrial MIS system compares directly with air knives, posing the industrial MIS system as an alternative for pre-sorting before high-performance but low-throughput material recovery options such as LIBS and XRF. For comparison, we also include the result of using the industrial MIS system with air knives repeated from Figure 9 as if a combined system consisting of a two-stage sorting process. These results are summarised in Table 3.

The air knives system removed the most wrought samples compared to the other approaches; however, mass purity was the lowest. The mass was the lowest as air knives failed to eject the nine larger pieces of wrought, which equated to 119 g. The industrial MIS system helped reduce the amount of cast samples to 53 (394 g) compared to the 76 (575 g) from air knives. However, this system exhibited inferior performance in general. The industrial MIS system extracted the heavier wrought pieces, which the air knives dropped. The combined air knives and industrial MIS system showed more promise, returning the highest purity rates in the sample numbers (52.94%) and mass (49.2%).

### 4.3. Potential Factors of Reduced Industrial Performance

The results presented on the industrial MIS system did not produce the recovery and purity rates expected compared to the static MetalID system. To investigate this reduced performance, 43 selected pieces of wrought aluminium were passed through the industrial MIS system, resulting in the ejection of 26 pieces and 17 pieces allowed to drop. The 17 pieces were then placed back through the system, with the expectation that they would drop again. However, eight of the pieces ejected, and nine dropped. The final nine pieces were again put through the system twice to see if they were ejected. On the first run, six pieces were ejected and three dropped. On the second run, three ejected and six dropped.

The metal pieces that passed over the conveyor multiple times can be seen in Figure 10, with the number of runs required to eject the samples shown. At least five pieces ejected on the second and third runs were long and thin. One possibility is that the long thin geometry led to pieces passing between two sensors, causing both sensor measurements to fail to reach the trigger threshold. Other pieces ejected during the 2nd and 3rd runs were small, which may not have generated a large enough secondary magnetic field to pass the threshold, especially if it passed some distance from the centre of the coil.

As the industrial MIS system’s performance did not meet the expectation of the static MetalID system, further work was performed to investigate the potential causes.

Our first discovery was that when a measurement is close to the threshold, which determines whether a metal is present, and the algorithm would classify it as wrought if so. A reason for the wrought classification is that the training samples which give a low-intensity measurement across all frequencies, are wrought. The algorithm would learn the low-intensity measurements as wrought, which leads to this misclassification. This is not an issue if a metal piece passes over the centre of the coil as the peak measurement is used, and for most pieces, the measurement will not be near the threshold. However, if a metal piece passes over multiple coils, then this could lead to two different classification results. For example, an algorithm might be correct if most of a cast sample passed over one coil and was classified as cast. But if a small section of the sample passed over another coil, which gave measurements around the threshold, a wrought classification would occur, triggering the air ejectors. This would lead to cast metal samples triggering the air ejectors and being pushed into the wrought bin. For testing, this would also imply that the algorithm classified the metal piece as wrought because the air ejectors were triggered.

The threshold could be further increased to mitigate this error, but this would lead to small pieces not being detected. Alternatively, if two sensors detect the same metal piece, the sensor with the largest measurement would be used to make the decision. The two sensors would need to be in communication, adding more complexity. In addition, communication between sensors reduces the modularity of the design, as the induction coils cannot be considered independent systems, especially if a master controller is needed to verify or override results. Alternatively, a camera could detect which sensors a metal piece passes over to help decide which sensor output to use; however, again, this comes with added complexity.

The static MetalID system was reused to understand how the orientation of metal pieces presented to the coil determines the classification result. The metal sample was measured and then flipped over to a different orientation. The first measurement was used for the “$\bar{\mathrm{flipped}}$” (not flipped) dataset, and the first and second measurements of the sample were used for the “flipped” dataset. An example of a measured wrought metal sample can be seen in Figure 11. Figure 12 shows the F1 score of the “flipped” and “$\bar{\mathrm{flipped}}$” datasets for all algorithms.

All algorithms were re-tested with the “flipped” dataset to determine the machine-learning algorithm and hyperparameters that gives the highest F1 score using stratified 10-fold cross-validation. F1 scores increased with the “flipped” dataset for all algorithms, notably KNN, which increased from 0.7565 to 0.8461. The improvement in the F1 score shows that, by increasing the number of training samples and surfaces measured, the algorithms performed better. The future training of algorithms would benefit from the training measurements of multiple surfaces of the same piece to increase the training samples. The increase for KNN and other algorithms could be related to similar measurements on both sides of the piece, though this was not true for all pieces. Similar measurements benefit KNN the most because, when it is used to predict a new measurement, it compares the new measurement to the closest measurements from the training dataset.

Figure 12 shows that the SVM algorithm achieved the highest average F1 score of 0.9395 compared to the other algorithms. The SVM algorithm achieved a 0.9456 F1 score on 3 out of the 10 data shuffles. Across the 384 measurements, only 19 were classified incorrectly, 11 wrought and 8 cast. All measurements that were incorrectly classified were consistent between the shuffles, with the exception of one wrought piece, which was incorrect for two shuffles and another wrought piece, which was only incorrect for one shuffle.

Of the 19 incorrectly classified measurements, only 4 samples were incorrect on both sides. For the other 11 measurements, the other orientation of the metal sample had a correct classification. The pieces incorrectly classified in only one orientation had different surfaces exposed to the sensors, resulting in different lift-off. The pieces were incorrectly classified on both sides had a similar surface regardless of the orientation.

The difference in results when the orientation of the pieces was changed shows that the same piece can have different classification results depending on the side exposed to the sensors. Creating the training dataset with the metal samples passed over the sensor multiple times is critical as the piece could have different surfaces, which the algorithm needs to learn. This development explains how the wrought metal on the industrial MIS system had different classification results on different runs. Ideally, we would measure every sample on all possible faces. However, this is difficult to achieve on a moving conveyor, especially when travelling at 2 m/s, as the metal pieces tend to move and roll. However, putting the metals used for training through the system multiple times will improve the available training data. The improvement will come from more measurements on different orientations for the given size of the metal samples available.

The choice of machine learning algorithm could be another factor that reduced the industrial MIS system performance. If the algorithm overfits the training data, it will not generalise well and give poor classification results on unseen data. To test this, we combined the ‘Industrial-twitch’ and ‘Industrial-air knives’ datasets to create a new dataset consisting of 216 pieces with a ratio of 43%/57% (wrought/cast). We refer to this new dataset as ‘Static-test and air knives combined’. The combination of the two datasets allowed for testing unseen measurements on the static MetalID system to evaluate the machine-learning models. The four algorithms that achieved the highest F1 score in Section 4.1 were chosen and trained on the flipped training data, and tested on the ‘Static-test and air knives combined’ dataset. The results are shown in Figure 13.

Figure 13 shows that ANN had the highest F1 score with the unseen data compared to the other algorithms. As the weights and biases of the neural network are randomly chosen, the average ANN result is lower than the best model. The average F1 score of ANN is 0.76, which is still higher than the next best algorithm, the SVM polynomial. We used an SVM with an RBF kernel for the industrial results, which had a lower F1 score than the SVM polynomial. The F1 score of the SVM RBF algorithms was lower by 0.097 compared to the best ANN algorithm. The lower result with the SVM on unseen data showed that the algorithm may have overfitted to the training dataset. The SVM algorithm’s lower performance compared to ANN suggests that the ANN algorithm may be a more promising choice for future work.

The confusion matrix of the best ANN model can be seen in Table 4. Table 4 shows that the algorithm obtained 36 cast and 33 wrought measurements wrong. From these incorrect measurements, only 14 cast and 10 wrought were from the same pieces, corresponding to 7 and 5 pieces incorrectly classified on both orientations. Therefore, the algorithm correctly classified the 22 cast and 23 wrought samples in one orientation. The difference in the classification of a sample depending on its orientation means that the F1 score, recovery and purity rates could have been larger or smaller if we decided only to use one orientation. Measuring more than one orientation is important for future work to account for the wide variation in surface characteristics when scrap metal is used.

## 5. Conclusions

Magnetic induction spectroscopy (MIS) offers a new approach to separating wrought aluminium from cast aluminium within a Twitch waste stream. Previous work highlighted that the conductive difference between cast and wrought aluminium could be leveraged to classify the metals using induction. This paper presents the results of wrought classification within Twitch using MIS on a laboratory system (static MetalID system) and on a full industrial material separation and recovery system (industrial MIS system) and thus represents the first industrial trial of our approach.

We demonstrate that MIS can classify wrought aluminium on a static MetalID system with an F1 score of 0.9222, with a recovery rate of 89.66% and a purity rate of 94.96%. However, the static MetalID system results were not reproduced on the industrial MIS system. The test samples used produced a recovery of 63.89% and a purity rate of 60%. Results further diminished with the declining prevalence of the wrought material when tested on raw Twitch metal samples or those pre-sorted by an air knives system.

Potential causes of the reduced performance of the industrial MIS system were examined, with three main areas identified. First, we found that, if the sensor had a low intensity measurement close to the threshold, which is used to determine whether the metal was present, the algorithm would classify the metal piece as wrought and trigger the air ejectors. If a piece of cast covered one sensor and part of an adjacent sensor, the small signal on the partial measurement could be sufficient for the adjacent sensor to misclassify, firing the air ejector associated with that sensor. The air ejector could potentially hit the cast piece into the further bin, or during testing, the ejector firing would be used as an indicator that the algorithm classified the piece as wrought. A potential mitigation would be to have the sensors communicate with each other or a camera to detect the distance between the sample centroid and sensor axis.

The second cause is how the metal sample is presented to the sensor. Of 192 samples measured using the static MetalID system, only four were found to misclassify regardless of which side consistently faced the sensor. The small number of remaining misclassifications yielded a correct or incorrect result dependent on which way they were orientated. This shows that the metal surface facing the sensor is essential and can result in different accuracies depending on how the sample is exposed to the sensor coils. The pieces for the training dataset for the machine learning model can also be passed multiple times to expose different surfaces to the sensor.

Finally, the SVM algorithm used on the industrial MIS system may not be the best choice. Using the static MetalID system, an ANN performed better when tested on unseen data and generalised well, whereas the SVM may have overfitted to the training data. Future work may benefit from using an ANN.

By using the spectra obtained with MIS, we can discriminate material according to conductivity contrasts while minimising the effect of shape variation. MIS has shown good performance when classifying wrought aluminium within a Twitch waste stream on the static MetalID system in the laboratory phase of our work. The demonstrated industrial MIS system, combined with a separator to create a full sorting solution, would allow a high-throughput and mid-cost system to remove wrought pieces. However, challenges still remain in replicating the results of our laboratory tests at an industrial scale, as revealed by the performance of our first industrial trial.

## Figures and Tables

**Figure 1 sensors-23-07837-f001:**
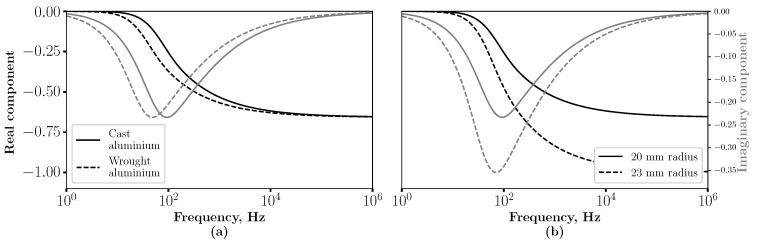
(**a**) The real and imaginary components of cast and wrought aluminium spheres of a 20 mm radius were calculated at a point 3 mm from the surface. (**b**) The real and imaginary components of a cast aluminium sphere with a radius of 20 and 23 mm calculated at a point 3 mm from the surface.

**Figure 2 sensors-23-07837-f002:**
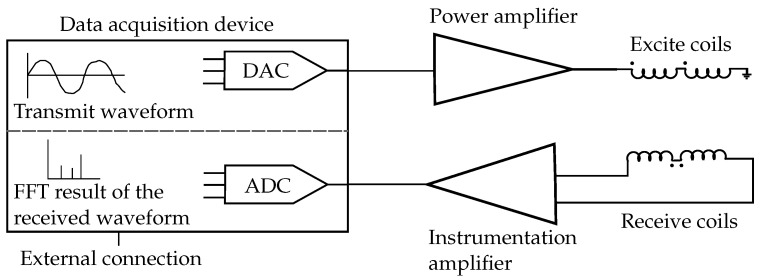
Schematic of analogue electronics of the MetalID system.

**Figure 3 sensors-23-07837-f003:**
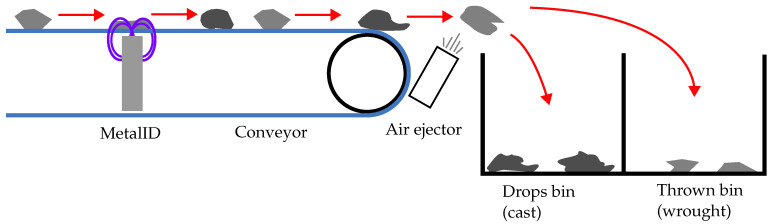
Diagram of the industrial MIS system, which consists of the MetalID sensors, conveyor and air ejectors.

**Figure 4 sensors-23-07837-f004:**
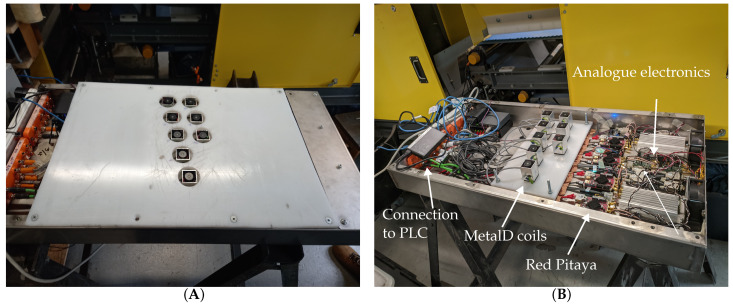
The box and cover used to house the electronics and MetalID coils (**A**). The contents of the housing box, which contains the MetalID coils, electronics and connection to the external PLC (**B**).

**Figure 5 sensors-23-07837-f005:**
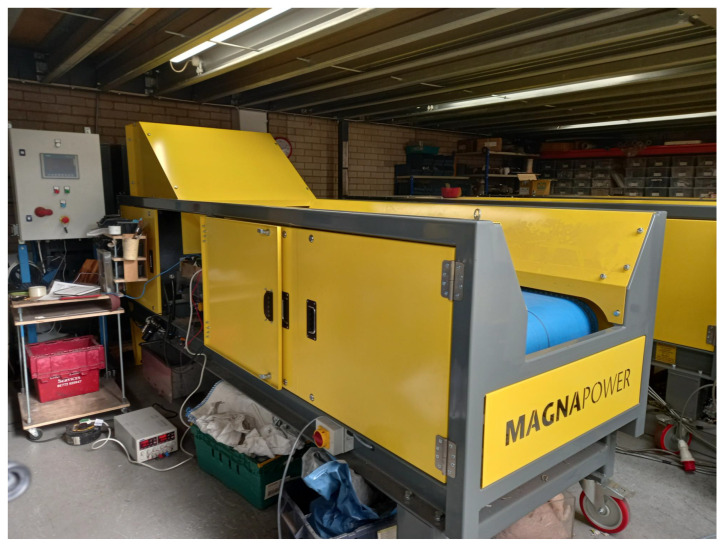
The industrial MIS system used for testing.

**Figure 6 sensors-23-07837-f006:**
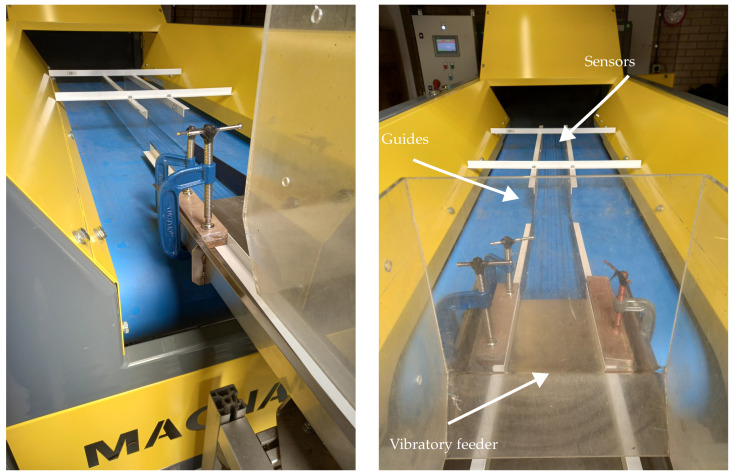
The vibratory feeder and guides used to guide the metal samples over the sensors.

**Figure 7 sensors-23-07837-f007:**
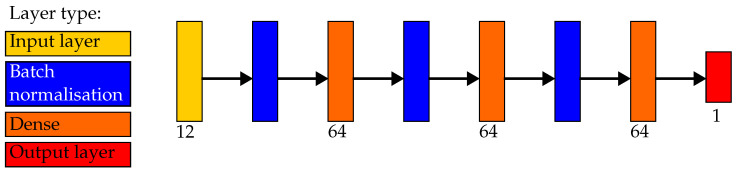
Architecture of the artificial neural network used in the reported results.

**Figure 8 sensors-23-07837-f008:**
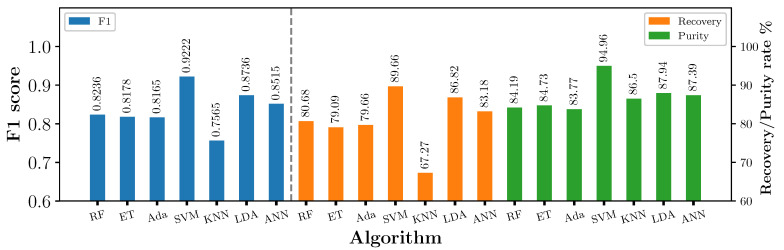
F1 score, recovery and purity rate of different algorithms using static MetalID system measurements.

**Figure 9 sensors-23-07837-f009:**
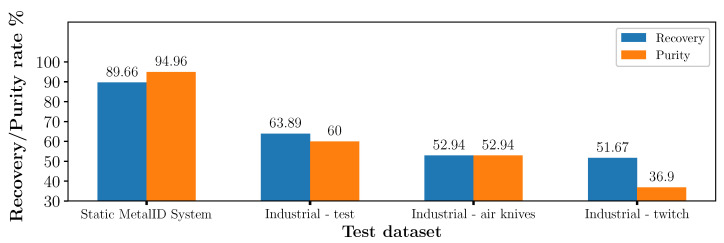
Recovery and purity rates on the industrial MIS system compared to static results.

**Figure 10 sensors-23-07837-f010:**
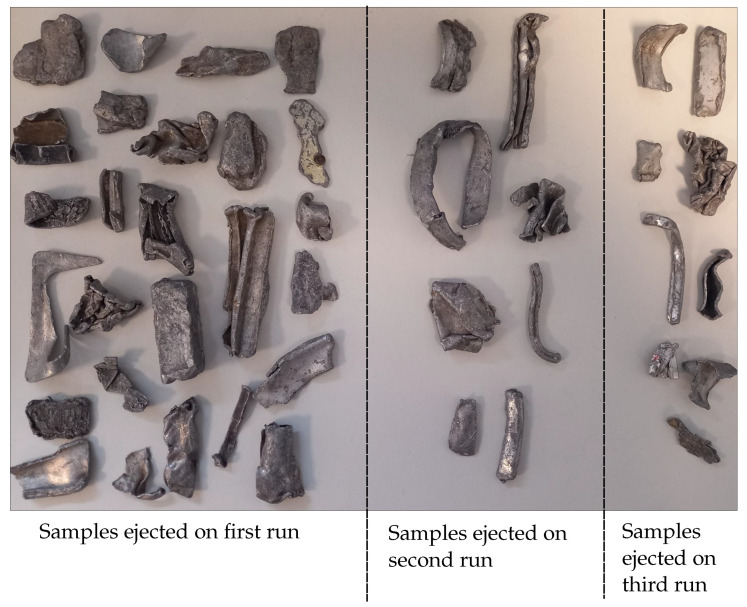
Wrought samples which were ejected on the first, second and third runs.

**Figure 11 sensors-23-07837-f011:**
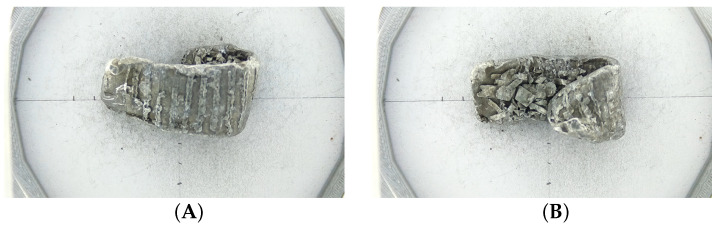
A sample which was measured (**A**), then flipped (**B**) and measured again. The flipped piece would present a different surface to the sensor.

**Figure 12 sensors-23-07837-f012:**
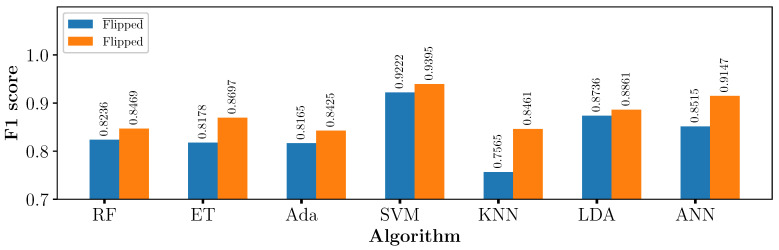
F1 score of pieces measured in one orientation compared to two orientations, using 10-fold cross-validation averaged across 10 random shuffles of the data.

**Figure 13 sensors-23-07837-f013:**
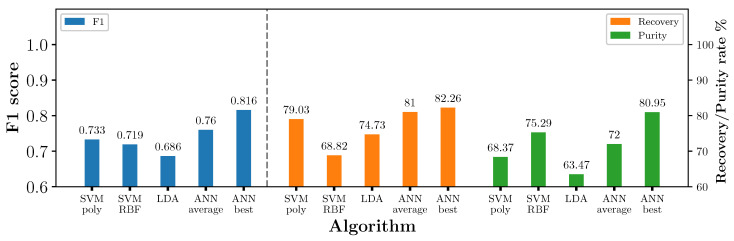
F1 score, recovery and purity rate of different algorithms tested on the unseen flipped test and air knives dataset.

**Table 1 sensors-23-07837-t001:** Cast and wrought aluminium samples used, which were sourced for a Twitch waste stream.

Section	Dataset	Wrought	Cast	Total	Ratio (%)
Section 4.1	Static-training	88	104	192	45/55
Section 4.2	Industrial-test	36	41	77	53/47
	Industrial-twitch	60 (380 g)	203 (2045 g)	263 (2425 g)	22/78 (15/85)
	Industrial-air knives	51 (261 g)	76 (575 g)	127 (836 g)	40/60 (31/69)

**Table 2 sensors-23-07837-t002:** Machine learning algorithms used and their hyperparameters that were varied.

Algorithm	Abbreviation	Hyperparameters
Random forest	RF	Number of estimators and decision trees
Extra randomised trees	ET	Number of estimators and decision trees
Adaboost	Ada	Number of estimators
Support vector machine	SVM	Kernel, regularisation
K-nearest neighbours	KNN	Number of neighbours
Linear discriminate analysis	LDA	N/A
Artificial neural network	ANN	Number of hidden layers

**Table 3 sensors-23-07837-t003:** Recovery and purity rates of wrought aluminium when different methods are used on a twitch waste stream.

Method	Sample Recovery (%)	Samples Purity (%)	Wrought Samples Ejected	Cast Samples Ejected	Mass Recovery (%)	Mass Purity (%)	Wrought Mass Ejected (g)	Cast Mass Ejected (g)
Industrial MIS	51.67	36.9	31	53	52.62	32.65	191	394
Air knives	85	40	51	76	68.68	31.22	261	575
Combination	52.94	52.94	27	24	32.37	49.2	123	127

**Table 4 sensors-23-07837-t004:** Confusion matrix for the model achieving the highest F1 score when predicting the “Static-test & air knives combined" dataset.

		Prediction		
		Cast	Wrought	Recovery rate (%)
Label	Cast	210	36	85.37
	Wrought	33	153	82.26
	Purity rate (%)	86.42	80.95	

## Data Availability

Not applicable.

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
