# Peer review of "Classification of Shredded Aluminium Scrap Metal Using Magnetic Induction Spectroscopy"

_sensors, 2023, doi:10.3390/s23187837_

Round 1
Reviewer 1 Report
The paper presents a very interesting experimental study on the use of magnetic induction spectroscopy for industrial aluminium scrap sorting.
In general, the paper is well written, technically sound and comprehensive. The results are well explained.
I only have a few minor comments.
1. The MetalID sensor system uses the excitation frequencies in the range from 3kHz to 60kHz. Looking at Fig. 1, it can be seen that both cast and wrought aluminium pieces feature imaginary (quadrature) response peaks below 100Hz. Have the authors considered using lower excitation frequencies so as to capture this characteristic part of the spectrum? This may help to better distinguish between the two materials.
2. Section 3.3. "Five datasets were used, which consisted of different ratios of wrought and cast aluminium..."
In Table I, only four types of datasets are given. Later, in Section 4.3 it becomes apparent that the fifth dataset is ’Static – test and air knives combined’. Perhaps this could be noted in Table I to avoid confusion.
3. In the proposed system only peak values of sensor signals (captured at multiple frequencies) have been used for classification. Have the authors considered using some other / additional features that may be extracted from the line scan, e.g. the spatial width of the response, or some other morphological features that might provide information on object size/shape?
Author Response
The paper presents a very interesting experimental study on the use of magnetic induction spectroscopy for industrial aluminium scrap sorting.
In general, the paper is well written, technically sound and comprehensive. The results are well explained.
I only have a few minor comments.
- The MetalID sensor system uses the excitation frequencies in the range from 3kHz to 60kHz. Looking at Fig. 1, it can be seen that both cast and wrought aluminium pieces feature imaginary (quadrature) response peaks below 100Hz. Have the authors considered using lower excitation frequencies so as to capture this characteristic part of the spectrum? This may help to better distinguish between the two materials.
Thank you for your comment. We have considered lower frequency, however, it is not practically feasible for this application. If we were to sample 100 Hz we would have a measurement every 10 ms rather than the 1.28 ms we currently obtain, resulting in less measurement per piece. Further, as the pieces move at speed on the conveyor (typically 2 m/s), they will have moved an appreciable distance with respect to their size over the time period of 1 cycle. We could run the conveyor belt slower. However, feedback from the industry has indicated that this would undesirably impact throughput and the economics of material recovery.
- Section 3.3. "Five datasets were used, which consisted of different ratios of wrought and cast aluminium..." In Table I, only four types of datasets are given. Later, in Section 4.3 it becomes apparent that the fifth dataset is ’Static – test and air knives combined’. Perhaps this could be noted in Table I to avoid confusion.
Thank you for your comment. We have updated the text to “four datasets”. The 4th dataset is a synthesis of 2 datasets we mention and use later in the paper. We did not want to be misleading and include it in Table 1 because it is not a unique dataset.
- In the proposed system only peak values of sensor signals (captured at multiple frequencies) have been used for classification. Have the authors considered using some other / additional features that may be extracted from the line scan, e.g. the spatial width of the response, or some other morphological features that might provide information on object size/shape?
This is an excellent comment and something we have considered, and we are in the process of researching. Previous work has used a camera to detect a sample and extract the colour, we plan to use a similar camera to extract shape features.
Reviewer 2 Report
The authors of this manuscript used magnetic induction method to separate wrought and cast aluminium. The topic is highly relevant to the industry and several tests were performed using both static and industrial systems. The manuscript is well-written, and figures are clearly presented. Before the publication, the authors should consider addressing a few minor revisions as follows.
1. Judging from the results, the industrial system effectively classified aluminium. The authors also compared the detection on sides and addressed the effect of the orientation, which are essential factors in real operations. Could the authors add the comments and references on how other methods or setups deal with the random placements and orientations of specimen?
2. The tests with 7 machine learning algorithms are commendable. It would be beneficial for the readers to add the reference regarding the performance of machine learning algorithms in other applications. The discussion is extensive but lacks of cited articles.
3. The authors should ensure format consistency. The abbreviations for algorithms should be identical for all figures. Examples of issues in the reference list are as follows.
The page range (or Article number) is missing from Ref [3].
The page range (or Article number) and the volume are missing from Ref [13]. If the article is not yet published, the doi or “in press” should be added.
The page range (or Article number) is missing from Ref [3].
“O’toole” in Ref [13] and [18] should be “O’Toole” as in [21] and [22].
In Ref [16], “no” should be omitted from “no pp. 149–159”.
Author Response
The authors of this manuscript used magnetic induction method to separate wrought and cast aluminium. The topic is highly relevant to the industry and several tests were performed using both static and industrial systems. The manuscript is well-written, and figures are clearly presented. Before the publication, the authors should consider addressing a few minor revisions as follows.
- 1. Judging from the results, the industrial system effectively classified aluminium. The authors also compared the detection on sides and addressed the effect of the orientation, which are essential factors in real operations. Could the authors add the comments and references on how other methods or setups deal with the random placements and orientations of specimen?
LIBS and XRF will employ cameras to target the beam towards a piece regardless of the placement of the piece. Other methods, like eddy current separators, will work regardless of the position of the sample. We found no research in scrap metal recycling discussing the position and orientation of the scrap metal. There was work on using a 3D camera to help target the metal samples, so the following has been added to highlight this:
[Introduction Page 2] “To improve LIBS, the addition of a 3D camera has been proposed to detect the sample and the optimal target flat and uncontaminated surfaces”.
- The tests with 7 machine learning algorithms are commendable. It would be beneficial for the readers to add the reference regarding the performance of machine learning algorithms in other applications. The discussion is extensive but lacks of cited articles.
Thank you for your comment. We have added 3 references of work performing different machine learning tasks which compared the algorithms used in our work:
[Machine learning, page 10]: “Other work has compared the classification results of traditional algorithms such as random forests and ANN for high-resolution prediction of building energy consumption, where it was reported that ANN performed only marginally better than the decision tree based algorithm [23]. Ren compared ANN and SVM and found that both algorithms had a 10% improvement on the F1 score in testing when the datasets were balanced, but ANN performed better when the dataset was not balanced [24]. When SVM, RF and ANN were compared to classify tree species from airborne hyperspectral APEX images, it was found ANN performed better than SVM and RF [25]. It was reported that the benefit of using SVM and RF was the quicker training time, and their simpler methods allowed easier repeatability to produce higher accuracies. [25]”
- The authors should ensure format consistency. The abbreviations for algorithms should be identical for all figures.
Thank you for your comment. I have updated Figure 12 to use the abbreviations of the algorithms for consistency.
Reviewer 3 Report
Recycling of various materials, in particular aluminum, is an important problem both economically and in terms of environmental protection. the authors of this paper presented an innovative method of selection and recovery using magnetic induction spectroscopy (MIS) to classify wrought from cast aluminium. The proposed method is innovative and very effective. The presented results are very interesting and important from the application point of view.
Author Response
Thank you for your kind and supportive words. We are pleased recognised the importance of the work and we look forward to continue our research in this area.